# Diagnostic Accuracy of Nasopharyngeal Swab Cultures in Children Less Than Five Years with Chronic Wet Cough

**DOI:** 10.3390/children8121161

**Published:** 2021-12-08

**Authors:** Ali A. Asseri, Nasim Khattab, Dima Ezmigna, Nabil J. Awadalla, Cori Daines, Wayne Morgan

**Affiliations:** 1Department of Pediatrics, King Khalid University, Guraiger, Abha 62529, Saudi Arabia; 2Department of Pediatrics, The University of Arizona, Tucson, AZ 85724, USA; nasimkhattab@peds.arizona.edu (N.K.); cdaines@arc.arizona.edu (C.D.); wjmorgan@arizona.edu (W.M.); 3Department of Pediatrics, University of Florida, Gainesville, FL 32611, USA; dezmigna@ufl.edu; 4Department of Family and Community Medicine, College of Medicine, King Khalid University, Abha 62529, Saudi Arabia; njgirgis@yahoo.co.uk; 5Community Medicine Department, Mansoura University, Mansoura 35516, Egypt

**Keywords:** bacterial pathogens, bronchoalveolar lavage (BAL), chronic wet cough, nasopharyngeal swab, persistent bacterial bronchitis (PBB)

## Abstract

Background: It is necessary to find a non-invasive and accurate procedure to predict persistent bacterial bronchitis (PBB) causative organisms and guide antibiotic therapy. The study objective was to compare the diagnostic accuracy of nasopharyngeal swab cultures with bronchoalveolar lavage (BAL) cultures in children with PBB. Methods: Nasopharyngeal swab and BAL fluid specimens were collected and cultured for bacterial pathogens prospectively from less than five-year-old children undergoing flexible bronchoscopy for chronic wet cough. Results: Of the 59 children included in the study, 26 (44.1%) patients had a positive BAL bacterial culture with neutrophilic inflammation. Prevalence of positive cultures for any of the four common respiratory pathogens implicated in PBB (*Moraxella catarrhalis*, *Streptococcus pneumoniae*, *Staphylococcus aureus*, and *Haemophilus influenzae*) was significantly higher (*p* = 0.001) in NP swabs compared to BAL fluids (86.4% and 44.1% of PBB cases, respectively). NP swab cultures for any of the four main bacterial pathogens had 85% (95% CI: 65–96%) and 48% (95% CI: 31–66%) sensitivity and specificity of detecting PBB, respectively. Positive and negative predictive values were 56% (95% CI: 47–65%) and 80% (95% CI: 60–91%), respectively. In conclusion, in children less than 5 years of age with chronic wet cough (PBB-clinical), a negative NP swab result reduces the likelihood of lower airway infection; however, a positive NP swab does not accurately predict the presence of lower airway pathogens. Flexible bronchoscopy should be considered in those with recurrent PBB-clinical or with clinical pointers of central airway anomalies.

## 1. Introduction

A pediatric cough that persists for longer than four weeks is considered chronic and mandates further investigations [1,2,3,4]. Chronic cough is one of the common worrisome complaints among children and leads to frequent medical consultations [5,6]. Although childhood asthma is considered first in the differential diagnosis of chronic cough, other serious diseases that cause irreversible lung damage should be ruled out, such as chronic suppurative lung disease (CSLD) and bronchiectasis [2,3,7,8,9]. Recently, persistent bacterial bronchitis (PBB) was considered one of the most common etiologies of chronic cough in children who otherwise looked healthy [1,10,11]. There is growing evidence that PBB may be a precursor of CSLD and bronchiectasis and should be treated to improve quality of life and avoid irreversible lung damage [5,8,12,13].

PBB, first described in 2006, is a major cause of chronic cough in children [10]. Although the exact prevalence of PBB among children with chronic cough is unknown, several studies have estimated that around 40% of children referred to pediatric pulmonology services to evaluate chronic cough had PBB [10,14,15]. The first definition of PBB required a bronchoscopy to identify the causative pathogen of bronchitis [10] and confirm the diagnosis. It was modified later to include clinical criteria and considered bronchoscopy to obtain bronchoalveolar lavage (BAL) cultures for persistent or recurrent cases. The most updated definition of PBB includes the following: presence of chronic (>4 weeks’ duration) wet or productive cough, absence of specific clinical pointers that suggest alternative causes of wet or productive cough (ineffective airway clearance disorders or aerodigestive disorders), and cough that resolved following a 2–4-week course of an appropriate oral antibiotic [9]. Regarding the causative bacteria, *H. influenzae* was the most common organism, found in 28–58% of children, with *Streptococcus pneumoniae* (13–58%) and *Moraxella catarrhalis* (17–59%) the two other most frequently detected organisms [2,7,10,11,16,17]. Recently, Hare et. al, reported that in children with chronic wet cough, neither nasopharyngeal nor oropharyngeal swabs, alone or in combination, reliably predicted lower airway infection. However, they enrolled patients with PBB (21%) and 79% with chronic long-standing suppurative lung diseases (bronchiectasis 71% and CSLD 8%). Given the high percentage of patients with CSLD and bronchiectasis, generalization of the conclusion to include PBB patients needs further study [17].

Even though flexible bronchoscopy is a relatively safe procedure with a low risk of major complications (about 2%), it is still an invasive procedure to perform, especially in children [18]. Therefore, a non-invasive and readily available procedure is needed to predict the causative organism of the PBB and guide antibiotic therapy. In this prospective study of children undergoing flexible bronchoscopy for chronic cough, we aimed to calculate the sensitivity, specificity, and positive and negative predictive values of the nasopharyngeal swab in those with BAL-culture-based PBB.

## 2. Materials and Methods

### 2.1. Study Design, Setting, and Population

A total of 59 children aged between the ages of 4 and 60 months undergoing flexible bronchoscopy for recurrent episodes of chronic wet cough were prospectively enrolled between November 2016 and December 2018. Eligible children were less than five years old with a history of chronic wet/productive cough (>8 weeks) that was nonresponsive to inhaled corticosteroids. Children were evaluated in the pediatric pulmonary clinic for chronic cough at the Banner University medical center, Tucson, Arizona, United States. Children with comorbidities such as congenital airway anomalies, known primary immunodeficiency, congenital heart disease, chronic lung disease of prematurity, chronic respiratory insufficiency, cystic fibrosis, and primary ciliary dyskinesia were all excluded. Written informed consent was obtained from all parents/guardians of the enrolled subjects, and the study was approved by the University of Arizona institutional review board committee (Protocol Number: 1511237902R002, approved by Univ. of Arizona IRB on 7 December 2015).

### 2.2. Definitions 

The modified clinical-based case definition of PBB (also termed PBB-clinical) includes the presence of chronic wet cough (>4 weeks), absence of symptoms or signs of other causes of wet or productive cough, and cough resolved following a 2-week course of an appropriate oral antibiotic (usually amoxicillin-clavulanate) [9]. The microbiologic-based case definition of PBB (also termed PBB-micro) includes the presence of chronic wet cough (>4 weeks), lower airway infection (recognized respiratory bacterial pathogens growing in sputum or at BAL at a density of a single bacterial species >=10^4^ colony-forming units/mL), and cough resolved following a 2-week course of an appropriate oral antibiotic (usually amoxicillin-clavulanate) [10]. In addition, airway neutrophilic inflammation was defined as the percentage of neutrophilic counts on the BAL fluid greater than 6.5 percent of the total BAL white cell counts [10].

### 2.3. Procedures and Laboratory Techniques 

#### 2.3.1. Bronchoscopy

All patients were clinically stable (normal vital signs and chest examination) at the time of bronchoscopy. After induction of anesthesia, a 2.8 mm (Olympus XP 160) pediatric flexible fiberoptic bronchoscope was introduced transnasally, and 0.5 mL of lidocaine of 2% and 1% was sprayed at the levels of vocal cords and carina, respectively. Suction was avoided until the bronchoscope was completely wedged in the sampled bronchus to minimize contamination with upper airway pathogens. BAL was performed in the most affected area (identified radiologically or endoscopically during the procedure). Sterile, nonbacteriostatic normal saline solution (1mL/kg to a maximum of 20 mL) was instilled into the abnormal bronchus, followed by immediate suctioning into a mucus trap. A 0.5 mL aliquot of BAL fluid was transferred to a cryovial containing 0.5 mL of concentrated STGGB for bacterial culture. BAL fluid was transferred immediately to the laboratory. BAL fluid was sent for analysis, including cell count and differential, cytology, Gram stain, quantitative bacterial cultures and sensitivities, and lipid-laden macrophages. The bronchoscopy was performed in accordance with the international pediatric bronchoscopy guidelines [19].

#### 2.3.2. Nasopharyngeal Swab

After the bronchoscopy procedure, a rayon-tipped swab was inserted into the nasopharynx through a single nostril until it reached the posterior pharynx. If resistance was encountered, the swab was removed, and an attempt was made to pass the swab through the other nostril. Once the swab reached the posterior pharyngeal wall, a gentle rotation was performed for 2–3 s [20]. The swab was then withdrawn and placed in a tube containing 1.0 mL of skim milk tryptone glucose glycerol broth (STGGB) and then transported immediately to the laboratory [21].

#### 2.3.3. Microbiology 

A Gram-stained smear of the specimen showed Gram-positive and Gram-negative cocci; a part of the specimen was inoculated onto Columbia blood agar, bile salt agar, and chocolate agar. The organisms were isolated after 48 h of incubation at 37 °C under 5% CO_2_, which was identified by microscopy (bench method) then followed by the Vitek 2 fully automated system for confirmation. Potential pathogens, including *S. aureus*, *Streptococcus pneumoniae*, non-typable *Haemophilus influenzae*, and *Moraxella catarrhalis* were confirmed by the Vitek 2 fully automated system.

### 2.4. Statistical Analysis

Statistical analyses were performed using Stata version 14 (StataCorp, College Station, TX, USA). BAL cell counts were not normally distributed, and thus non-parametric testing was used. Fisher exact tests and Mann–Whitney U-tests were used to study the differences between categorical and continuous variables, respectively. Categorical and continuous data were presented as proportions and median with interquartile range, respectively. A *p*-value of <0.05 was determined to be statistically significant.

## 3. Results

### 3.1. Patients’ Characteristics

The study included 59 patients with a chronic wet cough. Their ages ranged between 4 and 60 months, and their median (IQR) age was 22 (12–36) months. Most of them (57.6%) were males. Of the 59 children included in the study, 26 (44.1%) patients had positive BAL bacterial culture with neutrophilic inflammation, whereas 33 (56%) patients had negative BAL bacterial culture. There were no statistically significant differences between children with or without BAL fluid quantitative cultures ≥10^4^ CFU/mL regarding sex and anthropometric measurements (Table 1).

### 3.2. Prevalence of Positive Nasopharyngeal and Lower Airway Respiratory Bacterial Pathogens

Detection of respiratory bacterial pathogens in the nasopharyngeal and BAL fluid (≥10^4^ CFU/mL) cultures is shown in Table 2. Prevalence of positive cultures for any of the four common respiratory pathogens implicated in PBB (*Moraxella catarrhalis*, *Streptococcus pneumoniae*, *Staphylococcus aureus*, and *Haemophilus influenzae*) was significantly higher (*p* = 0.001) in NP swabs compared to BAL fluids (86.4% and 44.1% of PBB cases, respectively). Similarly, *Moraxella catarrhalis* was significantly higher in NP swabs than BAL fluids (38.9% and 18.6%, respectively). All three main organisms (*H. influenzae*, *S. pneumoniae*, or *M. catarrhalis*) were positive in both BAL and NP cultures of five patients (8.4%). Similarly, *Streptococcus pneumonia* was positive simultaneously in both NP and BAL fluid cultures in five patients (8.4%).

### 3.3. Diagnostic Accuracy of Nasopharyngeal Swab Culture 

BAL and NP cultures were significantly correlated (r = 0.55, *p* = 0.012). About 84.6% (22 out of 26) of the children with positive NP had a lower airway infection diagnosed by quantitative colony counts of BAL fluid. Validation of NP swab cultures for any of the four main bacterial pathogens (*H. influenzae, S. pneumoniae, Staphylococcus aureus*, or *M. catarrhalis*) against BAL fluid culture as the gold standard for PBB diagnosis is presented in Table 3. NP swab cultures had 85% (95% CI: 65–96%) and 48% (95% CI: 31–66%) sensitivity and specificity of detecting PBB, respectively. NP swab cultures had 56% (95% CI: 47–65%) and 80% (95% CI: 60–91%) positive and negative predictive values, respectively (Table 3).

### 3.4. Relations of BAL Neutrophil Percentages with the Results of BAL and NP Swab Cultures

Figure 1 illustrates the median and interquartile range (IQR) of BAL neutrophil percentages in patients according to the results of BAL and NP swab cultures. The BAL neutrophil percentage was significantly (*p* = 0.0005) higher in patients with positive BAL culture than those with negative BAL culture. Similarly, BAL neutrophil percentage was significantly (*p* = 0.023) higher in patients with positive NP swab culture than those with negative NP swab culture.

The median BAL neutrophil percentage in children with BAL positive culture and negative NP was 49.0% with the IQR of 34.0–67.0%. Furthermore, in children with negative BAL cultures there was no statistically significant difference (*p* = 0.127) in BAL neutrophil percentage between children with positive and negative NP swab cultures (median (IQR) = 4.0% (1.0–25.5%) and 8.0% (3.5–38.0%), respectively) (data not shown).

## 4. Discussion 

In children with chronic wet/productive cough, quantitative bronchoalveolar lavage (BAL) culture is the gold-standard method for diagnosing lower airway infection [17]. This study simultaneously compared the bacterial pathogens from nasopharyngeal and lower airways of children less than five years old with a chronic wet cough. More than 80% of the children with upper airway pathogens cultured from their nasopharynx had a lower airway infection diagnosed by quantitative colony counts of BAL fluid reflecting a significant correlation between NP and BAL cultures. Upper airway cultures for *M. catarrhalis*, *H. influenzae*, and *S. pneumoniae* only predicted lower airway infection in 56% of the cases. However, negative NP bacterial cultures did not rule out lower airway infections, which were seen in 4 out of 26 patients with confirmed microbiological PBB. Our results show that in this population of children less than five years of age with chronic wet cough, NP swab cultures have a high negative predictive value (more than 80%), reflecting a low frequency of false-negative (high sensitivity).

Previous studies of young children less than five years old with chronic lower airway disorders have found similar results [17,22,23,24]. However, most of the published studies included patients with cystic fibrosis or established bronchiectasis [17,25,26,27,28]. Our study included children with no known preexisting chronic airway disorders. Therefore, the high negative prediction of NP swabs could be helpful as a non-invasive and simple procedure to screen for PBB in children less than five years old with a chronic wet cough. However, in cases of recurrent PBB or failure to respond to the antibiotics, flexible bronchoscopy is still the gold-standard procedure for confirming the diagnosis and ruling out other etiologies, including coexistent central airway anomalies; examples are tracheomalacia, laryngotracheomalacia, and bronchomalacia. These anomalies cause more persistent symptoms and inadequate response to the usual antibiotic therapies [29].

Airway neutrophilia and endobronchial infection are common in children with chronic wet cough [16]. Our study showed that children with confirmed microbiological-PBB had significant neutrophilic inflammation. In addition, patients with positive NP swabs had neutrophilic inflammation as well. However, four patients had negative NP swabs in spite of having neutrophilic airway inflammation and positive BAL bacterial culture.

The most commonly cultured pathogens (from sputum or BAL) in children with PBB are non-typeable *H. influenzae*, *S. pneumoniae*, and *M. catarrhalis*, which are also found in the early stages of bronchiectasis [10,11]. In our study, *Moraxella catarrhalis* (42%) and *Haemophilus influenzae* (35%), followed by *Streptococcus pneumoniae* (19%), were the most common pathogens cultured from the BAL. These findings are in agreement with several published studies [30,31,32]. *M. catarrhalis*, *S. pneumoniae*, and *H. influenzae* are known to colonize the normal upper airways [33] and could be a major source of lower respiratory tract pathogens. However, the exact mechanisms underlying the similarities between upper airway colonizing bacteria and PBB causative pathogens are still unknown. One of the proposed mechanisms, particularly during acute viral respiratory infections, is the recurrent aspiration of upper airway secretions [12]. In addition, the anatomy of the upper airways in children can facilitate repeated microaspiration of heavily bacteria-laden nasal secretions, leading to chronic lower airway infection [24,34].

Several studies reported different diagnostic methods that compare upper- and lower-airway bacteria in children with PBB, CF, chronic suppurative lung disease, and bronchiectasis [17,24,26,27,28,35,36,37]. These studies concluded that the nasopharyngeal swab is more predictive and reliable in diagnosing lower airway infection than the oropharyngeal method. However, the positive prediction was between 50% and 60%, and that should be interpreted carefully given the high rate of nasopharyngeal bacterial colonization among children [38]. On the other hand, the high negative prediction could help screen children with suspected PBB if flexible bronchoscopy is not feasible.

Our study has several limitations. First, our study’s limited sample size predisposes it to type II error, and our results should be interpreted as preliminary. Second, although our results were consistent with previous research, the lack of a control group (healthy children) may have influenced differences in upper airway colonization and the presence of a real bacterial pathogen.

## 5. Conclusions

In children less than five years of age with chronic wet cough (PBB-clinical), a negative NP swab result reduces the likelihood of lower airway infection; however, a positive NP swab does not accurately predict the presence of lower airway pathogens. Therefore, non-invasive and reliable methods for diagnosing chronic lower airway infection in children with chronic wet cough are required to guide antibiotic choices. In addition, flexible bronchoscopy should be considered in those with recurrent PBB-clinical or with clinical pointers of central airway anomalies.

## Figures and Tables

**Figure 1 children-08-01161-f001:**
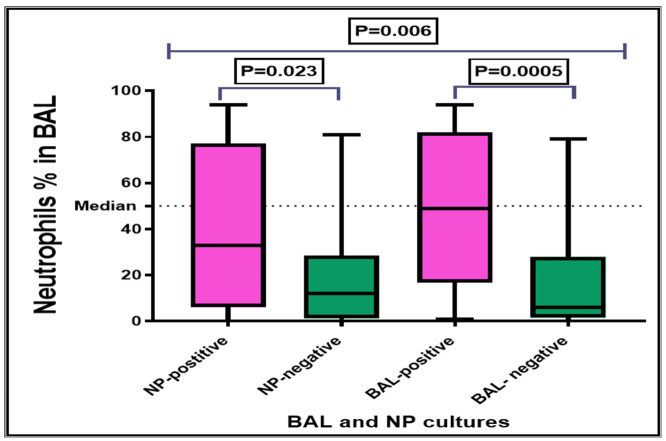
Median and interquartile range (IQR) of BAL neutrophil percentages in patients according to the results of BAL and NP swab cultures.

**Table 1 children-08-01161-t001:** Patients’ characteristics (*n* = 59).

Variables	Total	BAL Fluids ≥ 10^4^ CFU/mL		*p*-Value
		Yes *n* = 26	No *n* = 33	
Age in months *	22(12–36)	18.75 (12.7–30)	24 (12–37)	0.19
Sex, male %	34 (57.6)	15 (57.7)	19 (57.7)	0.99
Height (cm) *	83 (73–92)	81 (70–86.4)	89 (79–97)	0.15
Weight (kg) *	11.2 (9.2–14.3)	11.15 (9.2–13)	11.6 (9.39–15.1)	0.52

* Median (interquartile range); BAL, bronchoalveolar lavage; CFU, colony-forming units.

**Table 2 children-08-01161-t002:** Prevalence of positive nasopharyngeal and lower airway respiratory bacterial pathogens in children with clinical persistent bacterial bronchitis (*n* = 59).

Respiratory Pathogens	NP Swab Positive *n* (%)	Positive BAL Fluid ≥ 10^4^ CFU/mL *n* (%)	*p*-Value
*Moraxella catarrhalis*	23 (38.9)	11 (18.6)	0.01
*Haemophilus influenzae*	17 (28.8)	9 (15.2)	0.07
*Streptococcus pneumoniae*	5 (8.4)	5 (8.4)	0.99
*Staphylococcus aureus*	6 (10.1)	1 (1.7)	0.09
All the three main organisms (*H. influenzae*, *S. pneumoniae*, or *M. catarrhalis*)	5 (8.4)	5 (8.4)	0.99
Any of the four organisms	51 (86.4)	26 (44.1)	0.001

BAL, bronchoalveolar lavage; CFU, colony-forming units; NP, nasopharyngeal.

**Table 3 children-08-01161-t003:** Sensitivity, specificity, positive predictive value, and negative predictive value of nasopharyngeal swab for the diagnosis of persistent bacterial bronchitis in children < 5 years.

Positive NP Swab Cultures for Any of the Four Organisms	Persistent Bacterial Bronchitis (Bacterial Growth ≥ 10^4^ CFU/mL in BAL)		Sen. (95%CI)	Sp. (95%CI)	PPV. (95%CI)	NPV. (95%CI)
	Yes *n* = 26	No *n* = 33	22/26 = 85% (65–96)	16/33 = 48% (31–66)	22/39 = 56% (47–65)	16/20 = 80% (60–91)
Positive	22	17				
Negative	4	16				

NP = nasopharyngeal swab, Sen. = sensitivity, Sp. = specificity, PPV = positive predictive value, NPV = negative predictive value, CI = confidence interval.

## Data Availability

The datasets used in this study are available from the corresponding authors upon request.

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
