# Peer review of "Diagnostic Accuracy of Nasopharyngeal Swab Cultures in Children Less Than Five Years with Chronic Wet Cough"

_children, 2021, doi:10.3390/children8121161_

Round 1

Reviewer 1 Report

This a nice article about the possible  utility of nasopharyngeal swab compared to bronchoalveolar lavage culture in children with  persistent bacterial bronchitis. The authors has used a solid methodological approach and the paper is well written. The only limitation is the reduced sample size.

My only comment is related to this sentence: "Similarly, BAL neutrophils percent was significantly higher in patients with positive NP swab culture than those with negative NP swab culture". Can this effect be due to those patients with positive NP swab and positive BAL culture? Please, analyze separately the relation between BAL neutrophils and positive NP swab in those patients with negative BAL culture. 

Author Response

The authors are much grateful to the reviewer for the constructive comments and suggestions on our manuscript. Kindly find the attached point-by-point response to the comments.

Reviewer 2 Report

This paper is very interesting and well written. All results are clearly presented and I have only some minor comments: 

  1. Definitions: I suggest to underline what kind of definitions the authors have used in this paper, PBB major or PBB minor?
  2. The authors are briefly discussing other methods of comparing upper and lower airway bacteria flora and refer to some papers using oropharyngeal swabs. However, some of these papers are old (Ref nr 26 and 27). It would have been interesting if the authors also could include a discussion whether laryngeal aspirate, a non-invasive method frequently used in children with CF, PCD and PBB, can be used. 

Author Response

(The authors gave the same response as above.)
